# Cuproptosis genes in predicting the occurrence of allergic rhinitis and pharmacological treatment

Ting Yi ⓘ*

Southern University of Science and Technology Hospital, Shenzhen, Guangdong, China

* Eating0712@163.com

## Abstract

### Background

While drug therapy and allergen immunotherapy are useful for alleviating symptoms of seasonal allergic rhinitis (AR), existing therapeutic options remain limited. Cuproptosis is a novel form of programmed cell death, and its role in allergic rhinitis has not yet been explored. Researching the interaction between cuproptosis and allergic rhinitis will likely pave the way for future treatment of this disease.

### Methods

A microarray dataset of AR patients and normal controls (GSE43523) were obtained from the Gene Expression Omnibus (GEO) database for differential gene analysis. Cuproptosis related genes were extracted from the differentially expressed genes (DEGs) to form the AR/cuprotosis-gene set and analyzed by the GO and KEGG databases. Intersection analysis further defined the AR signature genes (AR-sg). Consensus cluster analyses were used to define the AR/cuprotosis-genes into subsets. Finally, AR signature genes were used as targets for drug prediction and molecular docking to identify candidate drugs that may affect SAR.

### Results

Four AR signature genes (*MRPS30*, *CLPX*, *MRPL13*, and *MRPL53*) were selected by the MCC, EPC, BottleNeck, and Closeness algorithms. Correlation analysis of the AR signature genes and immune genes showed strong interactions; xCell analysis identified multiple immune cell types and supported these cells' importance in the AR pathogenesis. Finally, drug target analysis suggests that 1,5-isoquinolinediol and gefitinib have the potential to become future AR treatments.

### Conclusion

Our study analyzed allergic rhinitis and cuproptosis related genes by the bioinformatics approach and predicted 1,5-isoquinolinediol and gefitinib as potentially useful drugs for treating AR patients in the future.

**Citation:** Yi T (2025) Cuproptosis genes in predicting the occurrence of allergic rhinitis and pharmacological treatment. PLoS ONE 20(2): e0318511. https://doi.org/10.1371/journal.pone.0318511

**Data Availability Statement:** All relevant data are within the paper and its Supporting Information files.

**Funding:** The author(s) received no specific funding for this work.

# 1 Introduction

Allergic rhinitis (AR) is a global health concern, affecting approximately 10–20% of the world's population [1, 2]. It is a recurrent type I allergy involving the nasal mucosa and includes symptoms such as runny and itchy nose, sneezing, and nasal congestion [3–5]. Allergic rhinitis can cause daytime drowsiness, irritability, depression, and affect patients' quality of life by disrupting attention, learning, and memory [6]. Upward 16% of US adult respondents suffered from an inability to fall asleep or insomnia almost every day because of AR in any one year [7, 8]. There are two main types of allergic rhinitis, seasonal (SAR) and perennial (PAR), and outdoor pollen is the main trigger [9]. However, the underlying pathogenesis of AR remains unclear, and more than 65% are related to genetic factors [10, 11]. Clinical management includes intranasal corticosteroids, leukotriene receptor antagonists, antihistamines, Omalizumab, and allergen immunotherapy (AIT). Existing studies have shown efficacy for intranasal corticosteroids in the treatment of SAR, whereas oral antihistamines are not significantly effective [12]. Intranasal corticosteroids alone rather than in combination with leukotriene receptor antagonists are also recommended for patients over 15 years of age [13]. The anti-IgE monoclonal antibody, Omalizumab, improves AR symptoms by eliminating mast cell differentiation and reducing histamine production, but symptom relief may take several weeks [14]. Unlike the aforementioned symptomatic reliefs, AIT reduces SAR symptoms and drug use by attempting to treat the root cause of the condition. However, patients find it challenging to stick with the treatment because it can take up to three years and is expensive [15, 16]. Thus, AR drug treatment options are limited, and existing therapies are suboptimal for some patients and often fail to meet patient expectations.

Apart from well-known programmed cell death mechanisms such as apoptosis and ferroptosis, recent studies identified a novel form of cell death, copper-dependent cell death (cuproptosis). Copper ion ($Cu^{2+}$) was found to bind directly to the fatty acylated parts of the mitochondrial tricarboxylic acid (TCA) cycle, resulting in the accumulation of fatty acylated proteins and reduction of iron-sulfur cluster proteins, which causes a proteotoxic stress response and eventually cell death [17, 18]. $Cu^{2+}$ is an essential metal element for human development and survival [19]; the homeostatic regulation of which is mainly located in the mitochondria [20]. Allergic rhinitis primarily involves risk genes related to the immune system, while cuproptosis is a new form of cell death caused by copper ions, although both are related to physiological processes within the body, they seem to be in different biological pathways in current research. The copper-zinc superoxide dismutases (CuZnSOD) formed by $Cu^{2+}$ are significantly associated with decreased oxidative activity in the airways and a decrease in CuZnSOD marks apoptosis and shedding of airway epithelial cells [21]. Previous studies have found that the $Cu^{2+}$ and CuZnSOD in the blood of patients with allergic rhinitis are significantly higher than those in the control group. Understanding the association between diseases and cuproptosis genes will provide potential research directions and new options for clinical treatment [22].

This study investigates the potential association between AR and cuproptosis molecular pathways by bioinformatic analyses of publicly available databases, to establish theoretical reference and guidance for the discovery and innovation of clinical treatment options for AR. The graphical abstract of this study is shown in S1 Fig.

# 2 Data and methods

## 2.1 Microarray datasets and differential analysis

Microarray datasets from AR patients and healthy controls (GSE43523) were extracted from the GEO database. Differentially expressed genes (DEGs) between controls and AR patients

were analyzed using the limma package in R, as previously described [23]. DEGs were defined as genes with a fold change (FC) $\geq$ 1.3 and a p-value < 0.05. The dataset GSE118243 was employed as an independent validation set.

## 2.2 Cuproptosis and immune-related genes

Potential cuproptosis related genes (347, false discovery rate (FDR) < 0.05) were extracted from a previous publication of genome-wide screening by CRISPR-Cas9 knockout [17]. We entered the Immunology Database and Analysis Portal (ImmPort; https://www.immport.org/home), selected the gene list, downloaded the gene summary, then 1793 immune-related genes were obtained from the Immunology Database and Analysis Portal (ImmPort; https://www.immport.org/home) after removal of duplicates.

## 2.3 Functional annotation and pathway enrichment analysis

Biological functions of genes were inferred from Gene Ontology (GO) and Kyoto Encyclopedia of Genes and Genomes (KEGG) analysis using the ClusterProfiler software package. For multiple test corrections, the *P*-values were adjusted using the Benjamini-Hochberg method or FDR, with FDR < 0.05 being the threshold. To assess the potential function of the identified DEGs, a gene interaction network was constructed with GENEMANIA (http://genemania.org/search/) [24].

## 2.4 Gene set enrichment analysis (GSEA)

GSEA was implemented to elucidate the biological significance of AR signature genes. The reference data set, "c2.cp.kegg.v11.0.symbols" genome, was obtained from the Molecular Characterization Database (MSigDB, http://software.broadinstitute.org/gsea/msigdb) [25]. To achieve normalized enrichment scores for each analysis, 1,000 genome permutations were performed. *P* < 0.05 was considered as significant enrichment.

## 2.5 Immunoinfiltration analysis

To identify immune-related genes, we used the xCell database (http://xCell.ucsf.edu/) which contains the gene signature of 34 immune cell types identified from 10,808 genes [26]. Among these immune cell types, 21 were lymphoid cell types. The Mann-Whitney U test performed statistical analysis of immune cell signatures in AR and control samples, with *P* < 0.05 considered statistically significant.

## 2.6 Consensus cluster analysis

The expression profiles of AR and cuproptosis related genes were used for unsupervised clustering by the Consensus Cluster Plus software package (50 iterations and 80% resampling rate) using the consensus clustering method [27]. The optimal number of cluster was determined by evaluating the consensus matrix plots, consensus cumulative distribution function (CDF) plots, relative changes in the area under the CDF curve, and trace plots. Principal component analysis (PCA) was used to define differences in the expression of AR and cuproptosis related genes between the two subtypes. The ggplot2 software was used to draw the PCA plots.

## 2.7 Small molecule drug prediction

AR signature genes were analyzed with Enrichr to predict drugs that may target these pathways [28]. Enrichr is an online tool that extracts drug and target data from the Drug Characterization Database (DSigDB) [29], a web-based free repository of GSEA drugs and their target

genes. DSigDB currently contains data from 22,527 genomes, including 17,389 drugs and 19,531 genes. Drugs with $P < 0.05$ were considered to be significantly associated with the target gene.

## 2.8 Molecular docking simulation

Simulation of molecular docking was performed with thioacetamide. The 3D crystal structures of the respective proteins of the AR signature genes were downloaded from the RCSB Protein Data Bank (http://www.rscb.org/pdb/). The predicted drugs that may bind the AR signature proteins were downloaded from the National Library of Medicine PubChem (https://pubchem.ncbi.nlm.nih) and saved as a spatial data file format (.SDF). Automatic Docking Tool version 1.5.6 was used to simulate the interaction between the AR signature proteins and their respective ligands. The binding of the protein-ligand complexes in 2D and 3D was visualized using the Discovery Studio Visualization Tools 2016.Manuscript Formatting

# 3 Results

## 3.1 DEG identification and analysis

We analyzed the GEO data from GSE126307 to compare the differential gene expression patterns between seven AR patients and five normal controls. A total of 478 differentially expressed genes (DEGs) were obtained, of which 222 were up-regulated and 256 were down-regulated (**Fig 1A**), with the top 50 DEGs shown in the heatmap (**Fig 1B**).

## 3.2 Cuproptosis related gene analysis

Of the 478 identified DEGs in AR patients compared to controls, 10 overlapped with a previously published cuproptosis gene set of 347 genes [17]. These AR/cuprotosis-related genes

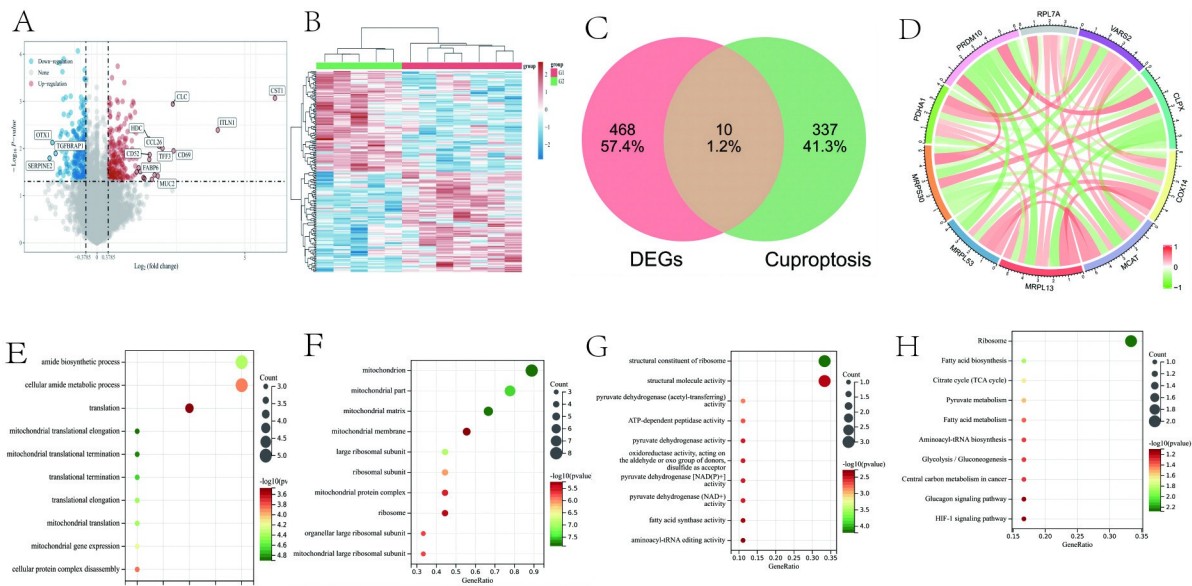

**Fig 1. Analysis of AR and cuproptosis-related genes.** (**A**) Volcano plot depicting the RNA expression levels of the differential genes between AR and control samples (red: up-regulated; blue: down-regulated; gray: no significant change). (**B**) Heatmap of the top 50 differentially expressed genes showing clustering of AR (red bar) and control (green bar) samples. (**C**) VENN diagram showing the intersection of AR differential genes and cuproptosis-related genes which is defined as the AR/cuproptosis-gene set. (**D**).

include seven upregulated (*VARS2*, *MCAT*, *RPL7A*, *MRPL13*, *PDHA1*, *MRPL53*, *COX14*) and three downregulated genes (*MRPS30*, *PRDM10*, *CLPX*; **Fig 1C**). Correlation analysis showed a strong correlation between these 10 AR/cuprotosis genes, by using the Pearson correlation coefficient, Its value is between -1 and 1, between 0 and 1 is positive correlated and we use the red color. (**Fig 1D**).

### 3.3 GO term and KEGG pathway analysis of AR/cuprotosis-related genes

Cuproptosis-related genes were analyzed by the GO and KEGG databases (**S1 Table**). Evaluation of GO terms showed that the biological processes that the AR/cuproptosis-genes are mainly involved in "amide biosynthetic process", "cellular amide metabolic process", "translation", "mitochondrial translational elongation" and "mitochondrial termination". The major cellular components that they are related to include "mitochondrion", "mitochondrial matrix", "mitochondrial part", "mitochondrial membrane", and "ribosomal large subunit". The AR/cuproptosis-genes have molecular functions such as "structural activity of ribosome", "pyruvate dehydrogenase (acetyl-transferrin) activity", "structural molecule activity", "ATP-dependent peptidase activity", and "pyruvate dehydrogenase" (**Table 1**). In the KEGG pathways

**Table 1. Gene ontology enrichment analysis for AR/cuproptosis-genes.**

| Category | GO ID | Term | Count | *P* value |
|---|---|---|---|---|
| Biological process | GO:0070125 | Mitochondrial translational elongation | 3 | <0.001 |
| | GO:0070126 | Mitochondrial translational termination | 3 | <0.001 |
| | GO:0006415 | Translational termination | 3 | <0.001 |
| | GO:0006414 | Translational elongation | 3 | <0.001 |
| | GO:0032543 | Mitochondrial translation | 3 | <0.001 |
| | GO:0043604 | Amide biosynthetic process | 5 | <0.001 |
| | GO:0140053 | Mitochondrial gene expression | 3 | <0.001 |
| | GO:0043603 | Cellular amide metabolic process | 5 | <0.001 |
| | GO:0043624 | Cellular protein complex disassembly | 3 | <0.001 |
| | GO:0006412 | Translation | 4 | <0.001 |
| Cell component | GO:0005739 | Mitochondrion | 8 | <0.001 |
| | GO:0005759 | Mitochondrial matrix | 6 | <0.001 |
| | GO:0044429 | Mitochondrial part | 7 | <0.001 |
| | GO:0015934 | Large ribosomal subunit | 4 | <0.001 |
| | GO:0044391 | Ribosomal subunit | 4 | <0.001 |
| | GO:0000315 | Organellar large ribosomal subunit | 3 | <0.001 |
| | GO:0005762 | Mitochondrial large ribosomal subunit | 3 | <0.001 |
| | GO:0098798 | Mitochondrial protein complex | 4 | <0.001 |
| | GO:0005840 | Ribosome | 4 | <0.001 |
| | GO:0031966 | Mitochondrial membrane | 5 | <0.001 |
| Molecular function | GO:0003735 | Structural constituent of ribosome | 3 | <0.001 |
| | GO:0004739 | Pyruvate dehydrogenase (acetyl-transferring) activity | 1 | 0.002 |
| | GO:0004176 | ATP-dependent peptidase activity | 1 | 0.002 |
| | GO:0004738 | Pyruvate dehydrogenase activity | 1 | 0.003 |
| | GO:0016624 | Oxidoreductase activity | 1 | 0.003 |
| | GO:0034603 | Pyruvate dehydrogenase [NAD(P)+] activity | 1 | 0.003 |
| | GO:0034604 | Pyruvate dehydrogenase (NAD+) activity | 1 | 0.003 |
| | GO:0005198 | Structural molecule activity | 3 | 0.004 |
| | GO:0004312 | Fatty acid synthase activity | 1 | 0.004 |
| | GO:0002161 | Aminoacyl-tRNA editing activity | 1 | 0.005 |

Table 2. Kyoto Encyclopedia of Genes and Genomes pathway analysis for SAR/cuproptosis-genes.

| | Term | Count | *p* Value |
|---|---|---|---|
| SAR/cuproptosis-genes | Ribosome | 2 | 0.005 |
| | Fatty acid biosynthesis | 1 | 0.014 |
| | Citrate cycle (TCA cycle) | 1 | 0.023 |
| | Pyruvate metabolism | 1 | 0.029 |
| | Fatty acid metabolism | 1 | 0.042 |
| | Aminoacyl-tRNA biosynthesis | 1 | 0.049 |

analysis, AR/cuproptosis-genes were enriched for "ribosome", "fatty acid biosynthesis", "citrate cycle (TCA cycle)", "pyruvate", "fatty acid metabolism", and "carbon metabolism" (**Table 2**).

### 3.4 Protein network analysis of AR/cuproptosis-related genes

To delve deeper into the possible functions of the AR/cuproptosis-related genes, we used Gene MANIA to draw the interaction network of the 10 AR/cuproptosis-genes and genes known to interact with them (**Fig 2A**), with relation-based multiple attributes of 20.02% co-expression and 79.98% physical interaction.

### 3.5 Selection of AR signature genes by MCC, EPC, BottleNeck, and Closeness algorithms

We used four different selection algorithms, MCC [30], EPC, Bottle Neck [31], and Closeness [32], to identify the best genes that can predict AR from the AR/cuproptosis-related genes. The

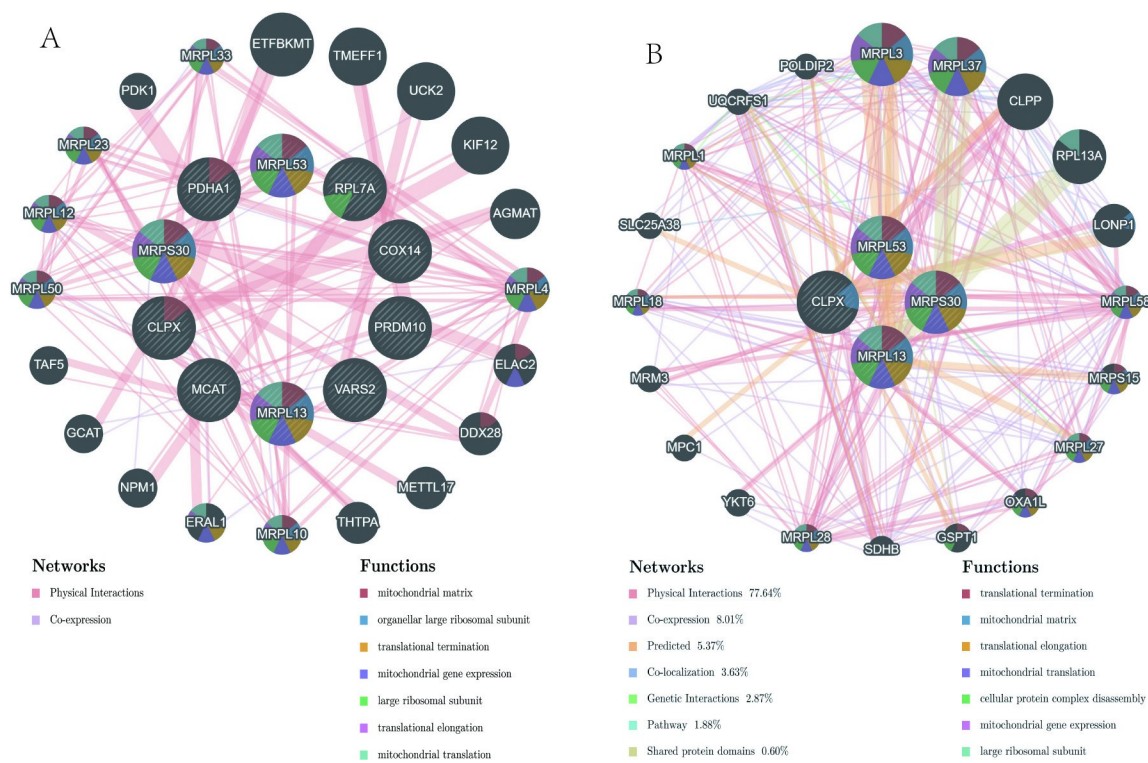

**Fig 2.** Protein interaction analysis for AR/cuproptosis-genes (**A**) and AR signature genes (**B**) using the GeneMANIA database. Genes are represented by the nodes, the colors of the nodes represent possible functions of corresponding genes, and the colored lines indicate type of network.

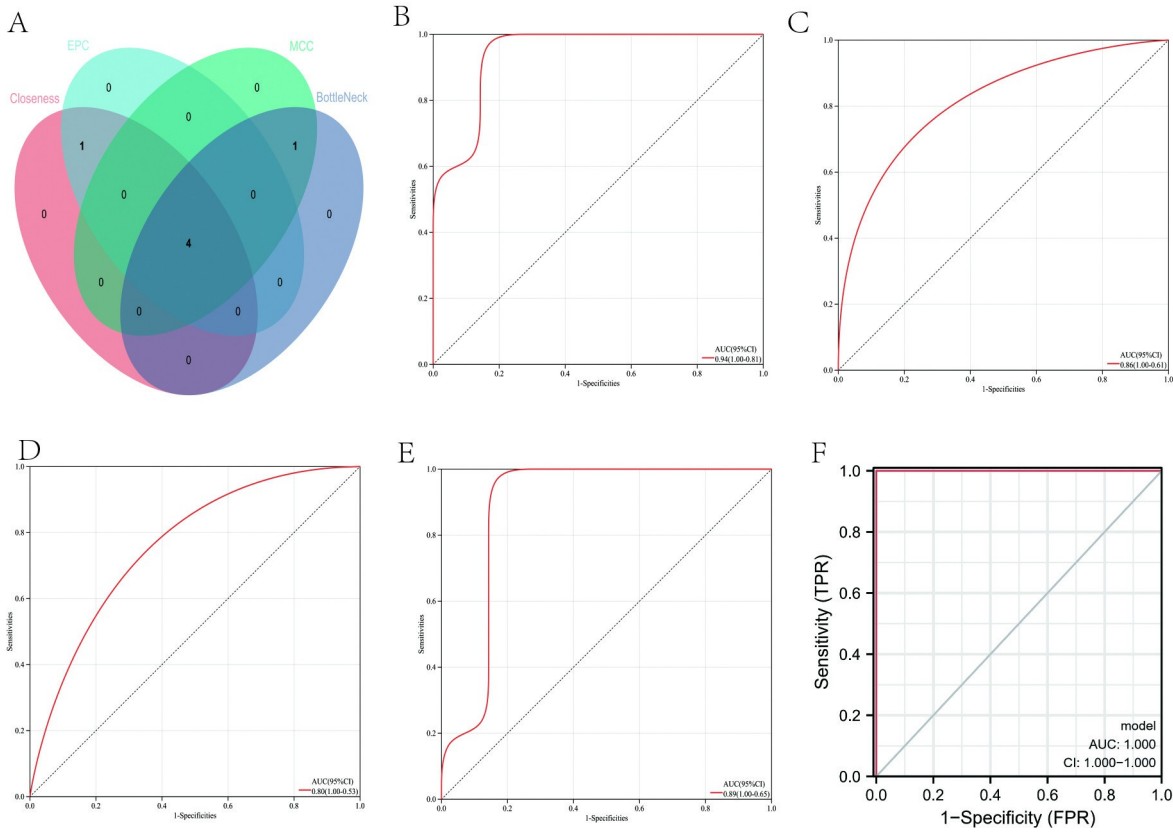

**Fig 3. AR signature genes and their diagnostic efficiency.** (**A**) VENN plot showing the intersection of potential signature genes selected by the MCC, EPC, BottleNeck, and Closeness algorithms. The four final AR signature genes are at the center of all four sets. Receiver operating characteristic (ROC) curves for individual signature genes: MRPS30 (**B**); CLPX (**C**); MRPL13 (**D**); MRPL53 (**E**) showing the predictive power of each gene for AR. All four signature genes were combined into one variable and analyzed by ROC curve (F) to estimate the diagnostic performance of the overall gene signature.

four algorithms identified five signature genes each: MCC (*MRPS30*, *CLPX*, *MCAT*, *MRPL13*, and *MRPL53*); EPC (*MRPS30*, *CLPX*, *MRPL13*, *MRPL53*, and *RPL7A*); Bottle Neck (*MRPS30*, *CLPX*, *MCAT*, *MRPL13*, and *MRPL53*); Closeness (*MRPS30*, *CLPX*, *MRPL13*, *MRPL53*, and *RPL7A*). These signature genes intersected to form the final four AR signature genes (SAR-sg): *MRPS30*, *CLPX*, *MRPL13*, and *MRPL53* (**Fig 3A**).

### 3.6 Diagnostic efficacy of AR signature genes in predicting AR

To determine whether the four AR-sg above are efficient at predicting AR, we first evaluated the diagnostic performance of each signature gene in predicting AR in the GSE43523 cohort. The area under the curve (AUC) values of the receiver operating characteristic (ROC) curve for each of the four AR-sg: *MRPS30* (0.94; **Fig 3B**), *CLPX* (0.86; **Fig 3C**), *MRPL13* (0.80; **Fig 3D**), and *MRPL53* (0.89; **Fig 3E**), demonstrate that these signature genes were able to predicate the occurrence of AR. Combining all four genes into one variable improved the AUC value to 1.0, indicating that the predictive value of the combined gene signature is high (**Fig 3F**).

Next, we evaluated this AR gene signature on an independent dataset(GSE118243). The AUC values for the four individual genes *MRPS30* (0.762), *CLPX* (0.667), *MRPL13* (0.714), and *MRPL53* (0.810) demonstrate that these signature genes can estimate the occurrence of AR well (**Fig 4A**). Further fitting of all four genes as one variable again improved the AUC value to 1.0 and supports good diagnostic efficiency for AR (**Fig 4B**).

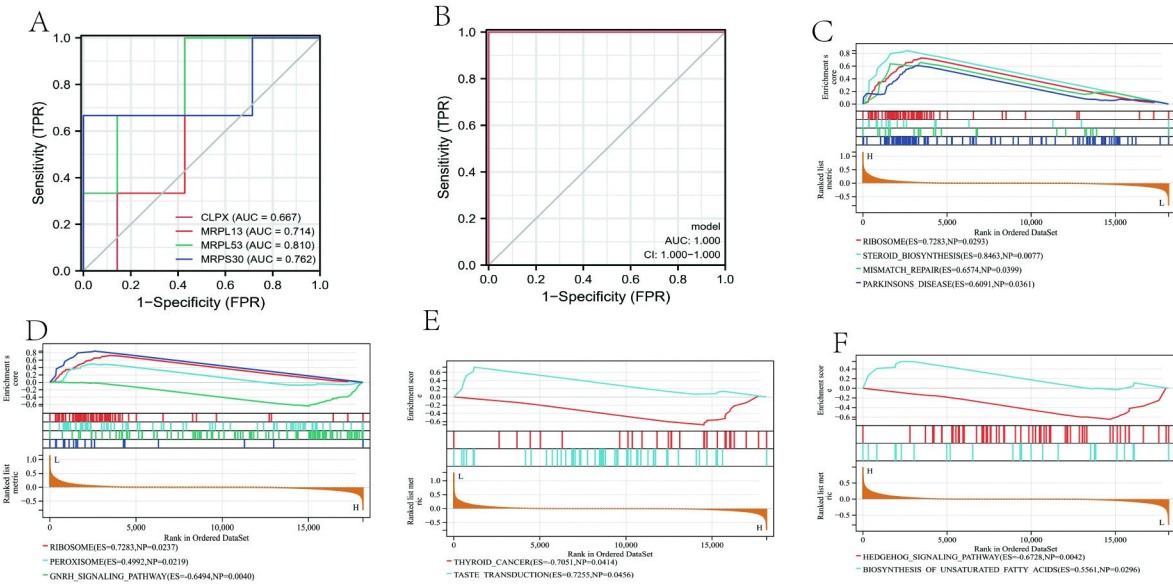

**Fig 4. Validation of AR signature genes and gene set enrichment analysis.** (**A**) ROC curve of individual signature genes validated in an independent data set, GSE118243. (**B**) ROC curve for the combined four gene signature validated on GSE118243. Pathway analysis for the four AR signature genes: *MRPL13* (**C**), *CLPX* (**D**), *MRPS30* (**E**), and *MRPL53* (**F**). Each graph showing signature genes can well estimate the occurrence of AR.

### 3.7 Protein interaction network analysis of AR signature genes

Similar to the interaction network for the AR/cuproptosis-related genes, we established an interaction network for the four AR-sg based on the functional annotation pattern of Gene MANIA (**Fig 2B**). In this case, the relation-based multiple attributes are 8.01% co-expression, 77.64% physical interaction, 5.37% prediction, 0.60% shared protein domain, 3.63% co-localization, and 1.88% pathway. Three of the four signature genes are highly correlated with mitochondrial gene expression (*P < 0.05*) and three are highly correlated with mitochondrial translation (*P < 0.05*).

### 3.8 Signaling pathway analysis of AR signature genes

To determine the likely pathways that the AR-sg may function in, we performed gene set enrichment analysis (GSEA). Our results showed that both *MRPL13* (**Fig 4C**) and *CLPX* (**Fig 4D**) are highly associated with the ribosome signaling pathway, while *MRPS30* is highly correlated with the thyroid cancer signaling pathway and showed high expression (**Fig 4E**). *MRPL53*, on the other hand, is highly associated with the hedgehog signaling pathway and showed low expression (**Fig 4F**).

### 3.9 Correlation analysis between AR-sg and immune genes and pathway enrichment analysis

The intersection of immune-related genes and differential genes was used to obtain 24 overlapping genes. (**Fig 5A**). Of these, 14 were highly expressed and 10 were lowly expressed. Most of the AR-related immune genes (AR/Imm) were expressed significantly higher in the AR group than in healthy controls, suggesting an increased immune response in AR patients. GO analysis of biological processes showed that AR/Imm-genes are mainly concentrated in the "regulation of cell communication", "regulation of signaling", "positive regulation of response to

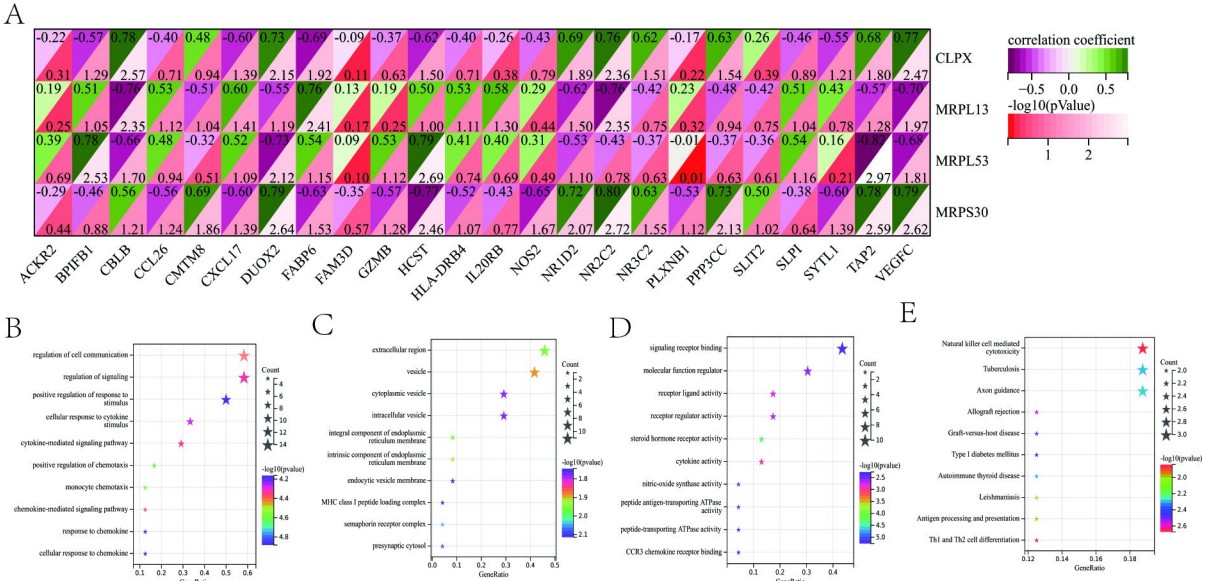

**Fig 5. Interaction between AR signature genes and immune related genes.** (**A**) Correlation analysis of the four AR signature genes and immune related genes. Visualization plot showing four characteristic genes are strongly correlated with immune-related genes. The AR/Imm genes were analyzed with GO terms for major biological processes (**B**), cellular components (**C**), and molecular functions (**D**), as well as KEGG pathways terms (**E**). The visualization plots show the top 10 terms for each analysis with distance along the x-axis indicating the count, color of stars indicating *p*-value, and size of stars indicating the count.

stimulus", "cellular response to cytokine signaling pathway" and "cytokine-mediated signaling pathway" (**Fig 5B**). The major cellular components that these genes are involved in include "extracellular regions", "vesicles", "cytoplasmic vesicles", and "intracellular vesicles" (**Fig 5C**). The main molecular functions that AR/Imm-genes contribute to include "signaling receptor binding", "molecular function regulator", "receptor ligand activity", and "receptor regulator activity" (**Fig 5D**). KEGG pathway analysis showed enrichment of AR/Imm-genes in "natural killer cell-mediated cytotoxicity", "tuberculosis", and "axon guidance" (**Fig 5E**).

## 3.10 Differential expression of immune cell signature in AR

Next, we evaluated the gene expression data with the xCell algorithm and calculated the scores for each sample based on the expression of genes related to 34 immune cell types (**Fig 6A**). Five immune cell types were significantly different between the AR and healthy control samples. Of these, four are related to bone marrow cells (mast T-cells, megakaryocyte-erythroid progenitor cells (MEPs), dendritic cells (DCs), and interstitial dendritic cells (iDCs)), and the remaining is for lymphocytes (CD4+_memory_T-cells). Elevation in the signature of these five immune cell types in the AR gene dataset suggests that immune cells may play a key role in the pathogenesis of AR.

## 3.11 Combined analysis of SAR/cuproptosis and immune-related genes identified two gene sets

Revisiting the 10 AR/cuproptosis-related genes, we found that they can be clustered into two groups, C1 and C2, based on consensus matrix plots (**Fig 7A**), consensus cumulative distribution function (CDF) plots (**Fig 7B**), relative changes in the area under the CDF curve (**Fig 7C**), and trace plots (**Fig 7D**). Principal components analysis (PCA) showed significant differences between the two groups (**Fig 7E**). Our additional heat map shows the distribution of 10 genes

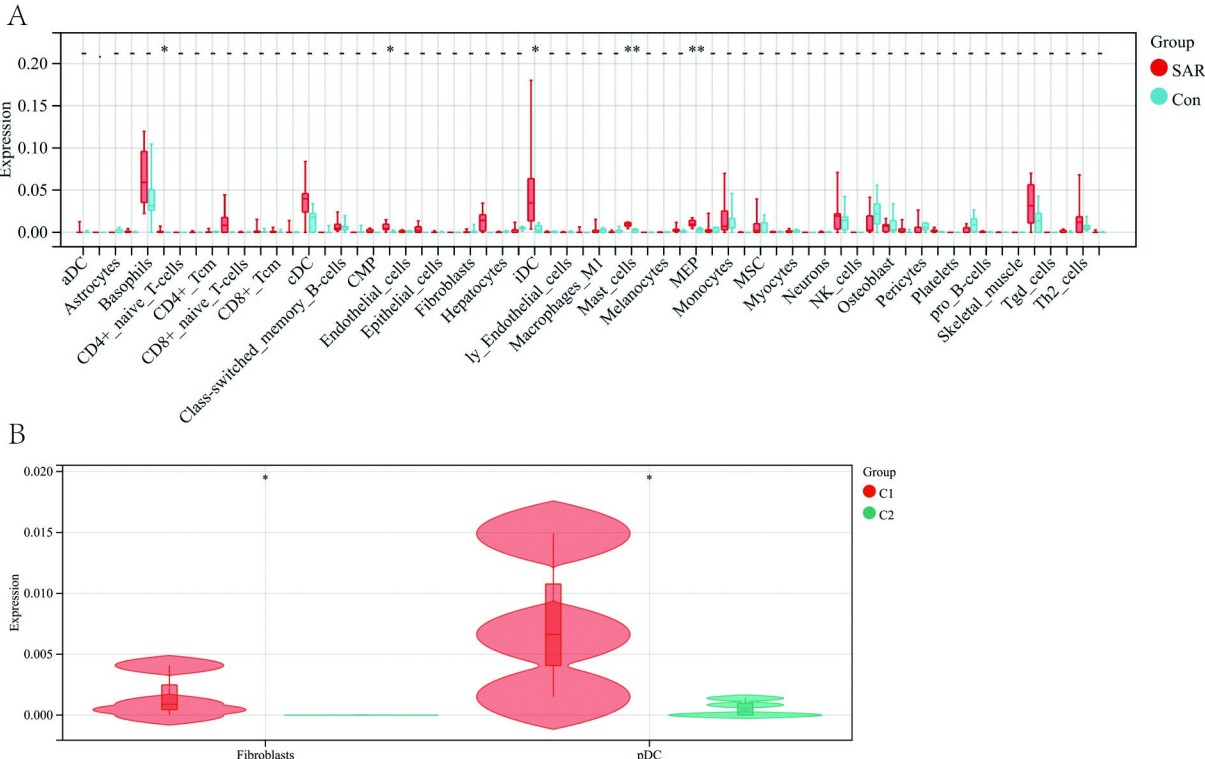

**Fig 6. xCell analysis of the AR/Imm and AR/cuproptosis-related genes.** (**A**) Graph comparing the expression of AR/Imm genes in each of the immune cell types defined in xCell. T indicate genes from AR patients and C indicate genes from healthy controls. (**B**) The 10 AR/cuproptosis-related genes were categorized into two groups, C1 and C2, based on consensus matrix analysis (see Fig 7), and subjected to xCell analysis. Graph showing the expression of fibroblast and pDC genes related to the two groups of AR/cuproptosis-genes.

in different subgroups(**S2 Fig**). We observed that both fibroblasts and plasmacytoid dendritic cells (pDCs) related genes expressed significantly higher C1 compared to C2 (**Fig 6B**). Thus, we defined C1 as the hyper-immune group and C2 as the non-immune group.

## 3.12 Drug prediction analysis and molecular docking

Finally, based on the data above we screened for candidates' drugs that may affect AR. Using the four AR-sg as drug targets, we used Enrichr to find drugs that are likely to target these genes. The results predicted 1,5-isoquinolinediol and gefitinib, with the highest odds ratio and combined score, as potential drugs for the treatment of AR patients (**Table 3**). To validate the above results, we performed in silico molecular docking simulations to determine whether the predicted small molecule drugs can bind well to the AR-sg target genes. The data showed multiple forces between gefitinib and *CLPX*, and hydrogen bonds can be formed with bond lengths of 4.23 Å (**Fig 7F**). These results suggest that the predicted drug candidates may achieve therapeutic effects in AR patients by inhibiting the expression of AR signature genes.

## 4 Discussion

Allergic reactions mainly involve abnormal activation of CD4 Th2 cells, secretion of interleukin-4 (IL-4), -5 (IL-5), -10 (IL-10), and -13 (IL-13), which increase the release of IgE from B lymphocytes into the blood [14]. In sensitized individuals, allergens deposited on the nasal mucosa bind to allergen-specific IgE on mast cells and trigger rapid release of the mediators

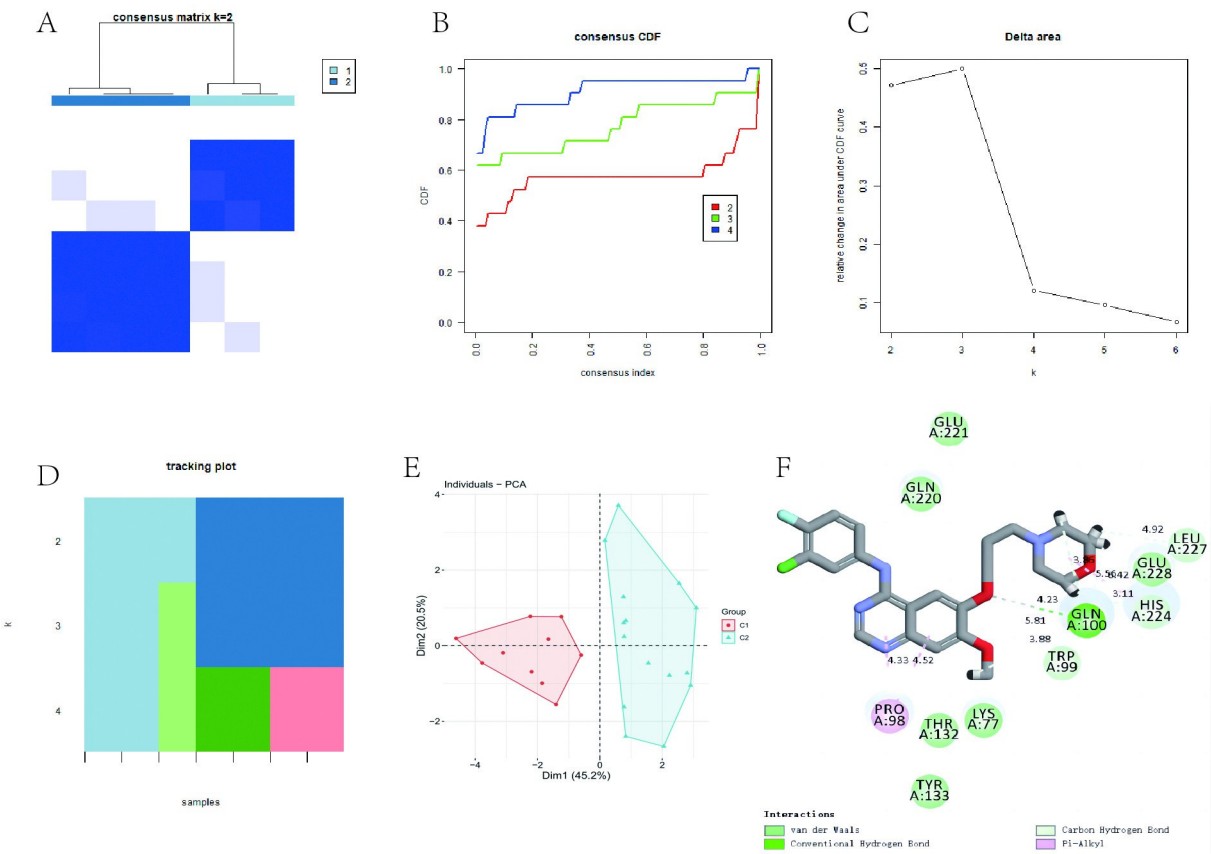

**Fig 7. Further analysis of AR/cuproptosis-related genes and drug interaction.** (**A**) Consensus matrix heatmap at k = 2 showing clustering of AR/cuproptosis-related genes into two subsets, C1 and C2. (**B**) Consensus cumulative distribution function (CDF) plot for k = 2–9. (**C**) Graph showing the relative change in the area under the CDF curve. (**D**) Tracing plot showing sample classification when k = 2–9. (**E**) PCA plots supporting the classification of AR/cuproptosis genes into two subsets, C1 and C2. (**F**) 2D image showing the docking complex formation between gefitinib and *CLPX*.

such as histamine, that cause the early symptoms of allergic rhinitis, including sneezing and nasal itchiness. In later phases of the allergic response, other inflammatory cells (basophils, eosinophils, and CD4+ T cells) infiltrate the allergy site when allergy mediators such as histamine, tumor necrosis factor alpha (TNFA), leukotriene C4 and prostaglandin D2 stimulate

**Table 3. Top 10 predicted drug candidates that may interact with the SAR signature genes.**

| Term | *P*-value | Odds Ratio | Combined Score |
| --- | --- | --- | --- |
| 1,5-isoquinolinediol HL60 DOWN | 0.014719032 | 90.97260274 | 383.7782872 |
| gefitinib HL60 DOWN | 0.01570767 | 85.11965812 | 353.5535367 |
| GNF-Pf-4325 CTD 00005408 | 0.017682713 | 75.40909091 | 304.2883339 |
| dirithromycin HL60 DOWN | 0.00445966 | 35.09386282 | 189.9519422 |
| irinotecan PC3 DOWN | 0.013980053 | 19.05616851 | 81.37219712 |
| doxorubicin PC3 DOWN | 0.014636037 | 18.58472086 | 78.50685079 |
| trichostatin A PC3 DOWN | 0.017819104 | 16.67992927 | 67.17815088 |
| alsterpaullone PC3 DOWN | 0.018428603 | 16.37271937 | 65.39020642 |
| chlorzoxazone HL60 DOWN | 0.022717631 | 14.57320872 | 55.15396901 |
| GW-8510 MCF7 DOWN | 0.022852898 | 14.52484472 | 54.88470168 |

endothelial cells to express adhesion molecules. This mass infiltration of immune cells contributes to nasal congestion [33–35]. In terms of signaling mechanisms, IL-6 is thought to regulate mast cell degranulation by activating the MAPK (mitogen-activated protein kinases)—STAT3 (activator of transcription 3) pathway in mast cell mitochondria [33], stimulating the electron transport chain and promoting OXPHOS (oxidative phosphorylation) -mediated ATP production [36–38]. In addition, IL-4 and -13 may activate several pathways including JAK/STAT, AKT/PI3K/mTOR, MAPK, and SRC to induce pro-inflammatory gene expression and effector function [39]. Allergic reactions primarily involve abnormal immune system responses, while cuproptosis is related principally to $Cu^{2+}$ metabolism and the abnormal dysfunction of mitochondria.

Currently, there are no studies on the mechanism of cuproptosis in allergic rhinitis. Thus, in the present study, we evaluated the differentially expressed genes between AR patients and controls for cuproptosis related genes and found 10 cuproptosis genes associated with AR (AR/cuproptosis genes), of which seven are up-regulated (*VARS2*, *MCAT*, *RPL7A*, *MRPL13*, *PDHA1*, *MRPL53*, *COX14*) and three are down-regulated (*MRPS30*, *PRDM10*, *CLPX*). To understand the roles of the cuproptosis-related genes in allergy rhinitis, we further performed GO enrichment analysis and newly found that the molecular function mainly involved in "structural constituent of ribosome", "pyruvate dehydrogenase (acetyl-transferring) activity", "ATP-dependent peptidase activity", "pyruvate dehydrogenase activity" and "oxidoreductase activity". Increasing ATP-dependent peptidase activity can promote mast cell degranulation. Targeted inhibition of cuproptosis may be a new direction for the treatment of AR.

KEGG pathway analysis found that AR/cuproptosis-genes were enriched for "ribosome", "fatty acid biosynthesis", "citrate cycle (TCA cycle)", "pyruvate", "fatty acid metabolism", and "carbon metabolism. The TCA cycle is a series of enzymatic reactions used by aerobic organisms to generate energy and involves the oxidation of acetate derived from carbohydrates, fats, or proteins. Succinate is an intermediate in this cycle and greatly enriched the allergy samples [40]. The study by Saude et al. showed that five metabolites acting in the TCA cycle (succinate, fumarate, oxaloacetate, cis-aconitate, and 2-oxoglutarate) were present at higher abundances in urine in allergy symptom who had recently suffered an exacerbation [41].

Further screening by four selection algorithms, MCC, EPC, BottleNeck, and Closeness, identified four signature genes (AR-sg: *MRPS30*, *CLPX*, *MRPL13*, and *MRPL53*) that showed superior performance in diagnosing AR, with AUC of 1.0 when all four genes were combined. This was validated in an independent data set. Analysis of the AR-sg signaling pathways and cross-correlation with immune gene expression revealed that the SAR-sg are mainly involved in natural killer cell-mediated cytotoxicity, tuberculosis, and axon guidance. Unfortunately, there is no current research on the relationship between the CLPX gene and AR.

*MRPS30* (mitochondrial ribosomal protein S30), also known as PDCD9 (programmed cell death protein 9), is a ribosomal protein in the mitochondria that regulates apoptosis [42]. *MRPS30* is highly expressed in vestibular schwannomas and triple-negative breast cancers and promotes tumor progression [43–45]. The long non-coding RNA divergent transcript of *MRPS30* can interact with RPS9, inhibit JAK-STAT signaling, and inhibit the progression of non-small cell lung cancer [46]. In allergic rhinitis, activation of the JAK/STAT signaling pathway by Th2 cytokines exacerbates the inflammatory response [47, 48]. Shen et al. found a clear association between JAK expression and increased AR susceptibility [49]. Animal studies have shown that JAK/STAT inhibitors can reduce the expression of OX40L (a thymic stromal lymphopoietin ligand) and the Th2 response, thereby improving the symptoms of allergic rhinitis [50].

*MRPL13* (mitochondrial ribosomal protein L13), as its name states, is also a ribosomal protein in the mitochondria. Mitochondria ribosomal proteins (MRPs) bind to rRNA in the

mitochondria to form mitochondrial ribosomes that translate mitochondria specific proteins [51]. Recent studies have revealed that many *MRPs* are associated with tumorigenesis [52]. Inhibition of *MRPL13* has been shown to precipitate OXPHOS dysfunction and enhance the invasiveness of hepatoma cells [53]. *MRPL13* knockdown can also decrease the expression of the vascular factor VEGF, which regulates endothelial cell growth and vascular permeability. VEGF is also considered an essential angiogenic factor that contributes to tumor development and chronic inflammation [54]. Animal studies have shown that VEGF increases vascular permeability by 10-fold and 106-fold over histamine [55]. About allergic responses, patients with AR or hyperresponsive airways showed significantly higher levels of VEGF and IL-5 mRNA compared to healthy controls [56]. We thus hypothesized that increased *MRPL13* expression in AR patients may lead to increased expression of VEGF, which aggravates the inflammatory response in AR patients. Potential signaling mechanisms of *MRPL13* involve PI3K, AKT, and mTOR as demonstrated in the regulation of BRCA tumor cell proliferation and migration [57]. This pathway also contributes to inflammation and pulmonary immune responses in allergic diseases [19]. Together these findings suggest that the increased inflammation in AR may be due to higher levels of *MRPL13* acting through the PI3K/AKT/mTOR pathway.

*CLPX* belongs to the AAA+ (ATP enzymes associated with various cellular activities) superfamily of ATP enzymes [58]. It is required for complex formation with caseinolytic peptidase P (*ClpP*) [59] to form ClpXP, which maintains proteostasis by degrading denatured or misfolded proteins [60–62]. ClpXP has a variety of substrates ranging from proteins that are involved in electron transport to those involved in metabolic processes and the tricarboxylic acid cycle (TCA cycle) [63], and those responsible for the integrity of the respiratory chain [64]. Mitochondrial *CLPX* can also function independently from *ClpP* to regulate heme biosynthesis in the mitochondria [65]. However, little is known about *CLPX* in allergic rhinitis. Similarly, the role of *MRPL53* in AR is also unclear. This protein is highly expressed in vocal and airway structures such as the tongue, throat, and trachea, as well as being present in skeletal muscles [66]. Like *MRPL13*, *MRPL53* is also a mitochondria ribosome protein and forms a large subunit of the mitochondrial ribosome, 39S. Therefore it has an integral role within the mitochondrial translation machinery and contributes to mitochondrial oxidative phosphorylation, which impacts on cellular functions such as cell growth, differentiation, and migration [67]. As both *CLPX* and *MRPL53* regulate fundamental cellular functions, they are likely to have some influence on the pathophysiology of allergic rhinitis. Thus, future investigations should focus on further understanding the contribution of these proteins to AR development.

To provide a potential application for the four AR signature genes that we have identified, we screened for drugs that may target these genes. 1,5-isoquinolinediol and gefitinib were predicted to bind the AR-sg and may be candidates for the treatment of AR. 1,5-isoquinolinediol, a poly (ADP-ribose) polymerase (PARP) inhibitor, has been shown to significantly inhibited inflammatory cell infiltration and the NF-κB signaling pathway [68, 69], The NF-κB signaling pathway is involved in activating dendritic cells (DCs) and is an inevitable pathway for the development of allergic rhinitis [70]. Inhibition of the NF-κB pathway can reduce the production and release of inflammatory cytokines and reduce infiltration of inflammatory cells into the submucosa, thereby reducing injury to the nasal mucosa and dampening allergic responses [71]. Thus, 1,5-isoquinolinediol may inhibit AR by modulating NF-κB signaling. Gefitinib inhibits the epidermal growth factor receptor (EGFR) and was developed as a cancer therapy, particularly for advanced non-small cell lung cancer [72]. A significant increase in EGFR mRNA expression can be seen in AR patients and contributes to nasal mucus hypersecretion [73], suggesting that gefitinib inhalers may be useful in reducing nasal mucosal inflammation.

Overall, our data suggest that cuproptosis genes may significantly impact the immune infiltration observed in allergic rhinitis. Thus, further studies on the causal relationship between

cuproptosis and SAR are needed in the future. In addition, we screened for potential therapeutic agents against the SAR signature genes with bioinformatics tools and provided a more cost-effective means of discovering new therapeutic strategies. Our data identified several targets that may be worth considering in the development of novel agents for the treatment of allergic rhinitis.

## Supporting information

**S1 Fig. The graphical abstract of this study.**
(TIF)

**S2 Fig. The possible correlations between the list genes.**
(TIF)

**S3 Fig. The heat map of the distribution of 10 genes in different subgroups.**
(TIF)

**S1 Table. The data of GO and KEGG analysis for SAR/cuproptosis-genes.**
(XLSX)

## Author Contributions

**Conceptualization:** Ting Yi.

**Data curation:** Ting Yi.

**Formal analysis:** Ting Yi.

**Methodology:** Ting Yi.

**Writing – original draft:** Ting Yi.

**Writing – review & editing:** Ting Yi.

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
