## [Decision Letter · Decision Letter 0]

28 Dec 2023

PONE-D-23-34994Cuproptosis Genes in Predicting the Occurrence of Allergic Rhinitis and Pharmacological TreatmentPLOS ONE

Dear Dr. Yi,

Thank you for submitting your manuscript to PLOS ONE. After careful consideration, we feel that it has merit but does not fully meet PLOS ONE’s publication criteria as it currently stands. Therefore, we invite you to submit a revised version of the manuscript that addresses the points raised during the review process.

We look forward to receiving your revised manuscript.

Kind regards,

Salman Sadullah Usmani, Ph.D.

Academic Editor

PLOS ONE

Journal Requirements:

Additional Editor Comments:

The manuscript requires revisions to address few major concerns, including the need for a more focused introduction, updated citations, improved presentation of results, and clearer explanations in the methodology. Additionally, specific attention should be given to figure quality, molecular docking result discussions, and establishing connections between cytokine responses and identified genes.

Reviewers' comments:

Reviewer's Responses to Questions

**Comments to the Author**

1. Is the manuscript technically sound, and do the data support the conclusions?

Reviewer #1: Yes

Reviewer #2: Partly

Reviewer #3: Partly

2. Has the statistical analysis been performed appropriately and rigorously? 

Reviewer #1: Yes

Reviewer #2: N/A

Reviewer #3: Yes

3. Have the authors made all data underlying the findings in their manuscript fully available?

Reviewer #1: Yes

Reviewer #2: No

Reviewer #3: Yes

4. Is the manuscript presented in an intelligible fashion and written in standard English?

Reviewer #1: Yes

Reviewer #2: No

Reviewer #3: Yes

5. Review Comments to the Author

Reviewer #1: Cuproptosis Genes in Predicting the Occurrence of Allergic Rhinitis and Pharmacological Treatment

The structure of the paper was well structured. But it is having following corrections

1. Proofread the entire manuscript.

2. Draw a graphical abstract to your study.

3. Compare your study with previous works.

4. Explain the novelty of the proposed approach.

Reviewer #2: In the current study, seasonal allergic rhinitis was investigated. Public microarray data were utilized for the analysis, including Gene Ontology (GO) and Kyoto Encyclopedia of Genes and Genomes (KEGG) pathway enrichment analyses. Several selection algorithms were employed to identify signature genes, resulting in the identification of four such genes. Additional validation datasets were used to establish predictive efficiency. Consensus cluster analyses were performed to further categorize the seasonal allergic rhinitis (SAR) or cuprotosis genes into subsets. Finally, the SAR signature genes were utilized as targets for drug prediction and molecular docking to identify candidate drugs that may impact SAR.

Several major concerns have been addressed and should improve in the revised version.

1. The introduction section is too general. It should be objective-oriented and highlight what bioinformatics applications have already been accomplished, what updates are necessary, and what has been applied in the current article.

2. The data used in this article were published in 2015, and since then, many research groups have utilized it in their studies. However, those articles were not cited, and the results were not compared.

3. Author should provide the results such as DE analysis, pathway enrichment analysis and others in the supplementary table.

4. The limitation mentioned, "Due to constraints in time and human resources, no further experimental verification has been conducted," is not satisfactory.

5. Overall, the writing is of low quality. The sentence structure lacks consistency.

6. The sentence and logic should be corrected as follows:

"Differentially expressed genes (DEGs) between controls and SAR patients were analyzed using the limma package in R, as previously described [25]. DEGs were defined as genes with a fold change (FC) ≥ 1.3 and a p-value < 0.05. The dataset GSE118243 was employed as an independent validation set."

Additionally, the citation [25] in this context may be inappropriate, as it doesn't provide sufficient information about the implications of the limma package. The original limma package should be cited for clarity and accuracy.

7. In the subsection "2.2 Cuproptosis and immune-related genes," the collection method for cuproptosis genes is mentioned, but crucial details, such as the rationale behind selecting this gene set and the objectives for its utilization, are missing.

8. Figure 1D: A heatmap illustrating the correlation would be clearer than a circus. The author mentions, "Correlation analysis showed a strong correlation between these 10 SAR/cuprotosis genes." However, the importance of this correlation analysis and the implications of the results are not discussed and should be addressed.

9. In the results section, it is reported that there are 10 overlapped genes between the 478 DEGs and the 347 cuproptosis genes. However, the statistical significance level of this overlap is not reported.

10. In the result section, only the terms of the pathways are reported for the GO and KEGG pathway analysis. However, there is a lack of discussion regarding the association of these pathways with Allergic Rhinitis.

11. Lines 265-276: The gene MRPS30 is primarily associated with breast cancer or tumors, with no direct relation to Allergic Rhinitis. The author attempts to establish a distant connection, which may be challenging to substantiate experimentally. The same comment applies to the other three signature genes.

Reviewer #3: The study by Ting Yi et al. explores the new territory of cuproptosis, a novel programmed cell death mechanism, in the context of seasonal allergic rhinitis (SAR). The study leverages a comprehensive bioinformatics approach, combining differential gene analysis, algorithm-based identification of SAR signature genes, and subsequent exploration of their interactions with immune genes. Furthermore, the research hunts into the potential therapeutic implications by identifying candidate drugs through molecular docking. While the study is very interesting and adds up the knowledge to the current literature and has the potential to be a very nice resource in the field, I did have a few concerns about the current version, as detailed below:

• Many figures suffer from poor quality, appear blurry and difficult to read, and need to change with high resolution ones.

• The molecular docking results should be presented and discussed with clear justifications for the selection of 1,5-isoquinolinediol and gefitinib as potential SAR treatments, considering factors like binding affinity and specificity.

• There is no apparent connection between the discussed paragraphs (line numbers 250-264) and the context of SAR and associated cytokine responses. There is a noticeable gap, and the authors have not adequately addressed the association of these responses with the identified genes. Proper transition is needed to bridge the gap between SAR, cytokines response and identified genes.

• It is advised that the authors should define abbreviations at least in the first instance. E.g. GEO in abstract, please check for other in entire manuscript.

• There are a lot of definite and indefinite articles missing along with some grammatical errors, also please correct the manuscript for undefined spaces.

• Number of spelling mistakes/typo errors.

• Citations and references must be in the correct and uniform format, Italics missing in some places.

6. PLOS authors have the option to publish the peer review history of their article (what does this mean?). If published, this will include your full peer review and any attached files.

Reviewer #1: No

Reviewer #2: No

Reviewer #3: No

---

## [Author Response · Author response to Decision Letter 0]

10 Feb 2024

Dear Editor and Reviewers:

Thank you for your letter and the constructive comments on this article in your busy schedule. I have carefully read the comments that you have given, and have discussed and revised each of these issues. The following is my list of revisions. In addition, I have resubmitted a new manuscript in the revised state, with the revisions highlighted in red. If there are any incorrect answers or questions in the manuscript, please do not hesitate to let me know.I hope the revised manuscript will accepted for publication in the journal of Plos One.

Yours sincerely

Ti Yi

Response to reviewers’ comments

Manuscript ID: PONE-D-23-34994

Title:Cuproptosis Genes in Predicting the Occurrence of Allergic Rhinitis and Pharmacological Treatment

Reviewer#1

The structure of the paper was well structured. But it is having following corrections

1. Proofread the entire manuscript.

Response:I sincerely thank the reviewer for careful reading.As suggested by the reviewer,I have proofread the whole manuscript again.

2. Draw a graphical abstract to your study.

Response:I sincerely appreciate the valuable comments.I have added a graphical abstract in the Figure S1.

3. Compare your study with previous works.

Response:Thanks for your question. Cuproptosis is a newly discovered cell apoptosis mode, which has been confirmed to play an important role in the occurrence of nasopharyngeal carcinoma(PMID: 36618352), head and neck tumor(PMID: 36344660), breast cancer(PMID: 37918452), lung cancer(PMID: 37037146), and other malignant tumors, and is closely associated with inflammatory bowel disease(PMID: 37090740), Crohn's disease(PMID: 36466876), rheumatoid arthritis(PMID: 35990673), and other non-neoplastic diseases. However, no studies have confirmed the association between Cuproptosis and allergic rhinitis.

4. Explain the novelty of the proposed approach.

Response:Thanks for your question. Because there have no studies confirmed the association between Cuproptosis and allergic rhinitis before. This study investigates the potential association between SAR and cuproptosis molecular pathways by bioinformatic analyses of publicly available databases, to establish theoretical reference and guidance for the discovery and innovation of clinical treatment options for SAR.

Reviewer#2

Several major concerns have been addressed and should improve in the revised version.

1. The introduction section is too general. It should be objective-oriented and highlight what bioinformatics applications have already been accomplished, what updates are necessary, and what has been applied in the current article.

Response:Thanks for your question.In order to make the research methods more accessible to readers, I have added a graphical abstract in the Figure S1. Through bioinformatics methods, I not only analyzed the correlation between copper death related genes and allergic rhinitis, but also carried out drug prediction.

2. The data used in this article were published in 2015, and since then, many research groups have utilized it in their studies. However, those articles were not cited, and the results were not compared.

Response:Thank you for your careful reading. Previous studies have explored the pathogenesis of allergic rhinitis and other related studies on the data set published in 2015. This paper mainly discusses the correlation between allergic rhinitis and cuproptosis, and there is no obvious correlation between the two.

3. Author should provide the results such as DE analysis, pathway enrichment analysis and others in the supplementary table.

Response:Thanks for your question. I have submit the result of DE analysis, pathway enrichment analysis in the supplementary table.

4. The limitation mentioned, "Due to constraints in time and human resources, no further experimental verification has been conducted," is not satisfactory.

Response:Thank you for your suggestions. Further verification by experiment does make the whole paper more convincing. As a clinician who just graduated with a master's degree, I completed the statistical analysis and writing of this paper through my own efforts. Due to the lack of resources, I do not have chance for further experiments. Therefore, thank you again for your valuable advice.

5. Overall, the writing is of low quality. The sentence structure lacks consistency.

Response:Thanks for your suggestion.Before submission,I have already invited a friend who is native English speaker from the USA to help polish my article.

6. The sentence and logic should be corrected as follows:

"Differentially expressed genes (DEGs) between controls and SAR patients were analyzed using the limma package in R, as previously described [25]. DEGs were defined as genes with a fold change (FC) ≥ 1.3 and a p-value < 0.05. The dataset GSE118243 was employed as an independent validation set."

Additionally, the citation [25] in this context may be inappropriate, as it doesn't provide sufficient information about the implications of the limma package. The original limma package should be cited for clarity and accuracy.

Response:Thank you for the comments. We show the revisions in red font in the manuscript.

7. In the subsection "2.2 Cuproptosis and immune-related genes," the collection method for cuproptosis genes is mentioned, but crucial details, such as the rationale behind selecting this gene set and the objectives for its utilization, are missing.

Response:There are 347 cuprotosis related genes in the present study. We intersected cuprotosis related genes with differentially expressed genes through bioinformatics, VENN diagram showing the intersection of SAR differential genes and cuproptosis-related genes which is defined as the SAR/cuproptosis gene set. (D) Correlation analysis of the overlapping SAR/cuproptosis genes. The SAR/cuproptosis genes were analyzed with GO terms for major biological processes (E)

8. Figure 1D: A heatmap illustrating the correlation would be clearer than a circus. The author mentions, "Correlation analysis showed a strong correlation between these 10 SAR/cuprotosis genes." However, the importance of this correlation analysis and the implications of the results are not discussed and should be addressed.

Response:Thanks for your question. Although the correlation coefficient matrix heat map is very intuitive, it can look very laborious and take up a lot of space. The use of circular form to display the correlation coefficient matrix makes reader feel soft and rich. Through using the pearson correlation coefficient,Its value is between -1 and 1, between 0 and 1is correlated and we use the red color.

9. In the results section, it is reported that there are 10 overlapped genes between the 478 DEGs and the 347 cuproptosis genes. However, the statistical significance level of this overlap is not reported.

Response:Thanks for your question. These ten SAR/cuprotosis-related genes include seven upregulated (VARS2, MCAT, RPL7A, MRPL13, PDHA1, MRPL53, COX14) and three downregulated genes (MRPS30, PRDM10, CLPX).The possible correlations between the SAR/cuprotosis-related genes is displayed on figureS2.

10. In the result section, only the terms of the pathways are reported for the GO and KEGG pathway analysis. However, there is a lack of discussion regarding the association of these pathways with Allergic Rhinitis.

Response:Thanks for your question. This pathways is important for Gene expression and transcription and I will further verify it in the later stage If the circumstances permit.

11. Lines 265-276: The gene MRPS30 is primarily associated with breast cancer or tumors, with no direct relation to Allergic Rhinitis. The author attempts to establish a distant connection, which may be challenging to substantiate experimentally. The same comment applies to the other three signature genes.

Response:Thank you for your suggestions. Further verification by experiment does make the whole paper more convincing. As a clinician who just graduated with a master's degree, I completed the statistical analysis and writing of this paper through my own efforts. Due to the lack of resources, I do not have chance for further experiments.

Reviewer #3: 

• Many figures suffer from poor quality, appear blurry and difficult to read, and need to change with high resolution ones.

Response:After I get the analysis results through R , I use the app“ AI” to combine them into graphs, and I have tried to make the pictures and figures clear to ensure readers' reading.

• The molecular docking results should be presented and discussed with clear justifications for the selection of 1,5-isoquinolinediol and gefitinib as potential SAR treatments, considering factors like binding affinity and specificity.

Response:Thank you for your suggestions.In line 250-264,the associated cytokine responses of SAR I described are common in allergic reactions.Thank you for your detailed reading.

• There is no apparent connection between the discussed paragraphs (line numbers 250-264) and the context of SAR and associated cytokine responses. There is a noticeable gap, and the authors have not adequately addressed the association of these responses with the identified genes. Proper transition is needed to bridge the gap between SAR, cytokines response and identified genes.

Response:Thank you for your suggestions.In line 250-264,the associated cytokine responses of SAR I described are common in allergic reactions.Thank you for your detailed reading.

• It is advised that the authors should define abbreviations at least in the first instance. E.g. GEO in abstract, please check for other in entire manuscript.

Response:Thank you for the comments. We show the revisions in red font in the manuscript.

• There are a lot of definite and indefinite articles missing along with some grammatical errors, also please correct the manuscript for undefined spaces.

Response:Thanks for your suggestion.Before submission,I have already invited a friend who is native English speaker from the USA to help polish my article.

• Number of spelling mistakes/typo errors.

Response:I feel sorry for my carelessness.In my resubmitted manuscript,the type has been revised.Thanks for your correction.

• Citations and references must be in the correct and uniform format, Italics missing in some places.

Response:Thanks for your careful checks.I borrowed the format of some paper previously published in the journal of “Plos One” and used the “Vancouver” style in reference and some fonts have been corrected to italics.Thanks for your correction.

---

## [Decision Letter · Decision Letter 1]

17 Apr 2024

PONE-D-23-34994R1Cuproptosis Genes in Predicting the Occurrence of Allergic Rhinitis and Pharmacological TreatmentPLOS ONE

Dear Dr. Yi,

Thank you for submitting your manuscript to PLOS ONE. After careful consideration, we feel that it has merit but does not fully meet PLOS ONE’s publication criteria as it currently stands. Therefore, we invite you to submit a revised version of the manuscript that addresses the points raised during the review process.

We look forward to receiving your revised manuscript.

Kind regards,

Salman Sadullah Usmani, Ph.D.

Academic Editor

PLOS ONE

Reviewers' comments:

Reviewer's Responses to Questions

**Comments to the Author**

1. If the authors have adequately addressed your comments raised in a previous round of review and you feel that this manuscript is now acceptable for publication, you may indicate that here to bypass the “Comments to the Author” section, enter your conflict of interest statement in the “Confidential to Editor” section, and submit your "Accept" recommendation.

Reviewer #2: (No Response)

Reviewer #3: All comments have been addressed

2. Is the manuscript technically sound, and do the data support the conclusions?

Reviewer #2: Partly

Reviewer #3: Yes

3. Has the statistical analysis been performed appropriately and rigorously? 

Reviewer #2: N/A

Reviewer #3: Yes

4. Have the authors made all data underlying the findings in their manuscript fully available?

Reviewer #2: No

Reviewer #3: (No Response)

5. Is the manuscript presented in an intelligible fashion and written in standard English?

Reviewer #2: No

Reviewer #3: Yes

6. Review Comments to the Author

Reviewer #2: The response from the author is unsatisfactory. The reviewer attempted to identify gaps and deficiencies so that the

authors could improve them, making the manuscript more understandable for readers upon publication.

However, it appears the author lacks the inclination to enhance the manuscript. I and reviewer 3 suggested

improvements to the writing. In response, the author stated, "Before submission, I had already enlisted the help

of a friend who is a native English speaker from the USA to assist in polishing my article."

However, having a native speaker from a different field revise a research article may not be sufficient.

Additionally, the discussion of the results is inadequate. I recommend discussing the GO and KEGG pathway

enrichment results to establish connections between the disease and the identified pathways.

I did not ask an experimental study for the GO and KEGG pathway.

The author's overall response likely avoids the reviewer's query

Reviewer #3: The authors have effectively considered and incorporated the concerns highlighted earlier. The revisions made by the authors have substantially improved the manuscript. I think, the modified version now appears well-prepared for publication.

7. PLOS authors have the option to publish the peer review history of their article (what does this mean?). If published, this will include your full peer review and any attached files.

Reviewer #2: No

Reviewer #3: No

---

## [Author Response · Author response to Decision Letter 1]

1 Jun 2024

Reviewer#2

I and reviewer 3 suggested

improvements to the writing. In response, the author stated, "Before submission, I had already enlisted the help of a friend who is a native English speaker from the USA to assist in polishing my article."However, having a native speaker from a different field revise a research article may not be sufficient.Additionally, the discussion of the results is inadequate. I recommend discussing the GO and KEGG pathway enrichment results to establish connections between the disease and the identified pathways. I did not ask an experimental study for the GO and KEGG pathway.The author's overall response likely avoids the reviewer's query

Response:I sincerely appreciate the valuable comments.I've added GO and KEGG results to the discussion section, and re-check the full text and make corresponding modifications

Reviewer #3:

The authors have effectively considered and incorporated the concerns highlighted earlier. The revisions made by the authors have substantially improved the manuscript. I think, the modified version now appears well-prepared for publication.

---

## [Decision Letter · Decision Letter 2]

6 Sep 2024

PONE-D-23-34994R2Cuproptosis Genes in Predicting the Occurrence of Allergic Rhinitis and Pharmacological TreatmentPLOS ONE

Dear Dr. Yi,

Thank you for submitting your manuscript to PLOS ONE. After careful consideration, we feel that it has merit but does not fully meet PLOS ONE’s publication criteria as it currently stands. Therefore, we invite you to submit a revised version of the manuscript that addresses the points raised during the review process.

We look forward to receiving your revised manuscript.

Kind regards,

Salman Sadullah Usmani, Ph.D.

Academic Editor

PLOS ONE

**Journal Requirements:**

Reviewers' comments:

Reviewer's Responses to Questions

**Comments to the Author**

1. If the authors have adequately addressed your comments raised in a previous round of review and you feel that this manuscript is now acceptable for publication, you may indicate that here to bypass the “Comments to the Author” section, enter your conflict of interest statement in the “Confidential to Editor” section, and submit your "Accept" recommendation.

Reviewer #2: (No Response)

Reviewer #4: (No Response)

2. Is the manuscript technically sound, and do the data support the conclusions?

Reviewer #2: No

Reviewer #4: Yes

3. Has the statistical analysis been performed appropriately and rigorously? 

Reviewer #2: N/A

Reviewer #4: Yes

4. Have the authors made all data underlying the findings in their manuscript fully available?

Reviewer #2: No

Reviewer #4: Yes

5. Is the manuscript presented in an intelligible fashion and written in standard English?

Reviewer #2: (No Response)

Reviewer #4: Yes

6. Review Comments to the Author

**Reviewer #2: **The response from the author to the first review was neither complete nor satisfactory. Most of my comments from the first review were ignored, which I also mentioned in the second review. If the author thoroughly addresses those comments and fills the gaps, I can consider the submission positively. Otherwise, I will have to reject it.

**Reviewer #4: **The authors have investigated the relationship between cuproptosis genes and allergic rhinitis (SAR) using bioinformatics analysis. It identifies four signature genes (MRPS30, CLPX, MRPL13, MRPL53) as potential biomarkers for SAR, with strong predictive accuracy. The study also suggests potential drugs (1,5-isoquinolinediol and gefitinib) for SAR treatment based on molecular docking analysis. The research contributes new insights into SAR pathogenesis and potential therapeutic targets. However, I have the following minor comments, before considering the manuscript for publication:

1. Rephrase the sentence in the abstract: "Cuproptosis is a novel mechanism of programmed cell death that has not been studied about allergic rhinitis" to improve clarity. Suggestion: "Cuproptosis is a novel form of programmed cell death, and its role in allergic rhinitis has not yet been explored."

2. In the introduction, add a clearer transition between the description of allergic rhinitis and the relevance of cuproptosis to strengthen the connection between the two topics.

3. In the results section, explain why combining the signature genes (MRPS30, CLPX, MRPL13, and MRPL53) significantly improves the AUC values in predicting SAR, emphasizing the added value of the combination.

4. Ensure that the versions of bioinformatics tools (such as R packages or Enrichr) are mentioned in the methods section for reproducibility and transparency.

5. In the discussion, elaborate on the connection between cuproptosis and immune responses in allergic rhinitis, offering a brief explanation of how cuproptosis might influence immune cells involved in the allergic reaction.

7. PLOS authors have the option to publish the peer review history of their article (what does this mean?). If published, this will include your full peer review and any attached files.

Reviewer #2: No

Reviewer #4: **Yes: **Sumeet Patiyal

---

## [Author Response · Author response to Decision Letter 2]

2 Nov 2024

Dear Editor and Reviewers:

Thank you for your letter and the constructive comments on this article in your busy schedule. I have carefully read the comments that you have given, and have discussed and revised each of these issues. The following is my list of revisions. In addition, I have resubmitted a new manuscript in the revised state, with the revisions highlighted in red. If there are any incorrect answers or questions in the manuscript, please do not hesitate to let me know.I hope the revised manuscript will accepted for publication in the journal of Plos One.

Yours sincerely

Ti Yi

Response to reviewers’ comments

Manuscript ID: PONE-D-23-34994R2

Title:Cuproptosis Genes in Predicting the Occurrence of Allergic Rhinitis and Pharmacological Treatment

Reviewer#2

The response from the author to the first review was neither complete nor satisfactory. Most of my comments from the first review were ignored, which I also mentioned in the second review. If the author thoroughly addresses those comments and fills the gaps, I can consider the submission positively. Otherwise, I will have to reject it.

Response: I apologize for my previous tardiness, and I especially want to express my sincere gratitude for your valuable advice,. I have rechecked the entire manuscript and compiled the GO and KEGG analysis results into table1 and table2.

Table 1. Gene ontology enrichment analysis for AR/cuproptosis-genes.

Category GO ID Term Count P value

Biological process GO:0070125 Mitochondrial translational elongation 3 <0.001

 GO:0070126 Mitochondrial translational termination 3 <0.001

 GO:0006415 Translational termination 3 <0.001

 GO:0006414 Translational elongation 3 <0.001

 GO:0032543 Mitochondrial translation 3 <0.001

 GO:0043604 Amide biosynthetic process 5 <0.001

 GO:0140053 Mitochondrial gene expression 3 <0.001

 GO:0043603 Cellular amide metabolic process 5 <0.001

 GO:0043624 Cellular protein complex disassembly 3 <0.001

 GO:0006412 Translation 4 <0.001

Cell component GO:0005739 Mitochondrion 8 <0.001

 GO:0005759 Mitochondrial matrix 6 <0.001

 GO:0044429 Mitochondrial part 7 <0.001

 GO:0015934 Large ribosomal subunit 4 <0.001

 GO:0044391 Ribosomal subunit 4 <0.001

 GO:0000315 Organellar large ribosomal subunit 3 <0.001

 GO:0005762 Mitochondrial large ribosomal subunit 3 <0.001

 GO:0098798 Mitochondrial protein complex 4 <0.001

 GO:0005840 Ribosome 4 <0.001

 GO:0031966 Mitochondrial membrane 5 <0.001

Molecular function GO:0003735 Structural constituent of ribosome 3 <0.001

 GO:0004739 Pyruvate dehydrogenase (acetyl-transferring) activity 1 0.002 

 GO:0004176 ATP-dependent peptidase activity 1 0.002 

 GO:0004738 Pyruvate dehydrogenase activity 1 0.003 

 GO:0016624 Oxidoreductase activity 1 0.003 

 GO:0034603 Pyruvate dehydrogenase [NAD(P)+] activity 1 0.003 

 GO:0034604 Pyruvate dehydrogenase (NAD+) activity 1 0.003 

 GO:0005198 Structural molecule activity 3 0.004 

 GO:0004312 Fatty acid synthase activity 1 0.004 

 GO:0002161 Aminoacyl-tRNA editing activity 1 0.005 

Table 2. Kyoto Encyclopedia of Genes and Genomes pathway analysis for SAR/cuproptosis-genes.

 Term Count p Value

SAR/cuproptosis-genes Ribosome 2 0.005 

 Fatty acid biosynthesis 1 0.014 

 Citrate cycle (TCA cycle) 1 0.023 

 Pyruvate metabolism 1 0.029 

 Fatty acid metabolism 1 0.042 

 Aminoacyl-tRNA biosynthesis 1 0.049 

Reviewer#4:

The authors have investigated the relationship between cuproptosis genes and allergic rhinitis (SAR) using bioinformatics analysis. It identifies four signature genes (MRPS30, CLPX, MRPL13, MRPL53) as potential biomarkers for SAR, with strong predictive accuracy. The study also suggests potential drugs (1,5-isoquinolinediol and gefitinib) for SAR treatment based on molecular docking analysis. The research contributes new insights into SAR pathogenesis and potential therapeutic targets. However, I have the following minor comments, before considering the manuscript for publication:

1.Rephrase the sentence in the abstract: "Cuproptosis is a novel mechanism of programmed cell death that has not been studied about allergic rhinitis" to improve clarity. Suggestion: "Cuproptosis is a novel form of programmed cell death, and its role in allergic rhinitis has not yet been explored."

Response:I sincerely appreciate the valuable comments.According to your advice, I have made relevant modifications , which you can find in the manuscript. See: Page1, line 10-11.

2.In the introduction, add a clearer transition between the description of allergic rhinitis and the relevance of cuproptosis to strengthen the connection between the two topics.

Response:Thanks for your valuable comments.According to your advice, I have made relevant modifications , which you can find in the manuscript. See: Page1, line 60-68.

3.In the results section, explain why combining the signature genes (MRPS30, CLPX, MRPL13, and MRPL53) significantly improves the AUC values in predicting SAR, emphasizing the added value of the combination.

Response:Thanks for your advice，with AUC of 1.0 when all four genes were combined. This was validated in an independent data set. Analysis of the AR-sg signaling pathways and cross-correlation with immune gene expression revealed that the SAR-sg are mainly involved in natural killer cell-mediated cytotoxicity, tuberculosis, and axon guidance.See: Page1, line 60-68.

4.Ensure that the versions of bioinformatics tools (such as R packages or Enrichr) are mentioned in the methods section for reproducibility and transparency.

Response:I’m sure that the versions of bioinformatics tools are mentioned in the methods section for reproducibility and transparency.

5.In the discussion, elaborate on the connection between cuproptosis and immune responses in allergic rhinitis, offering a brief explanation of how cuproptosis might influence immune cells involved in the allergic reaction.

Response:Thanks for your valuable comments.Allergic reactions mainly involve abnormal activation of CD4 Th2 cells, secretion of interleukin-4 (IL-4), -5 (IL-5), -10 (IL-10), and -13 (IL-13), which increase the release of IgE from B lymphocytes into the blood. Allergic reactions primarily involve abnormal immune system responses, while cuproptosis is related principally to Cu2+ metabolism and the abnormal dysfunction of mitochondria. Currently, there are no studies on the mechanism of cuproptosis in allergic rhinitis. Thus, in the present study, we evaluated the differentially expressed genes between AR patients and controls for cuproptosis related genes and found 10 cuproptosis genes associated with AR (AR/cuproptosis genes), of which seven are up-regulated (VARS2, MCAT, RPL7A, MRPL13, PDHA1, MRPL53, COX14) and three are down-regulated (MRPS30, PRDM10, CLPX). To understand the roles of the cuproptosis-related genes in allergy rhinitis, we further performed GO enrichment analysis and newly found that the molecular function mainly involved in “structural constituent of ribosome”, “pyruvate dehydrogenase (acetyl-transferring) activity”, “ATP-dependent peptidase activity”, “pyruvate dehydrogenase activity” and “oxidoreductase activity”. Increasing ATP-dependent peptidase activity can promote mast cell degranulation( doi.org/10.1016/j.jaci.2016.09.047). Targeted inhibition of cuproptosis may be a new direction for the treatment of AR. See: Page7, line 261-266.

---

## [Decision Letter · Decision Letter 3]

17 Jan 2025

Cuproptosis Genes in Predicting the Occurrence of Allergic Rhinitis and Pharmacological Treatment

PONE-D-23-34994R3

Dear Dr. Yi,

We’re pleased to inform you that your manuscript has been judged scientifically suitable for publication and will be formally accepted for publication once it meets all outstanding technical requirements.

Kind regards,

Vinay Kumar, Ph.D.

Academic Editor

PLOS ONE

Additional Editor Comments (optional):

Reviewers' comments:

Reviewer's Responses to Questions

**Comments to the Author**

1. If the authors have adequately addressed your comments raised in a previous round of review and you feel that this manuscript is now acceptable for publication, you may indicate that here to bypass the “Comments to the Author” section, enter your conflict of interest statement in the “Confidential to Editor” section, and submit your "Accept" recommendation.

Reviewer #2: (No Response)

2. Is the manuscript technically sound, and do the data support the conclusions?

Reviewer #2: No

3. Has the statistical analysis been performed appropriately and rigorously? 

Reviewer #2: N/A

4. Have the authors made all data underlying the findings in their manuscript fully available?

Reviewer #2: No

5. Is the manuscript presented in an intelligible fashion and written in standard English?

Reviewer #2: No

6. Review Comments to the Author

Reviewer #2: I appreciate the author's effort; however, I feel it does not fully resolve my key concerns, leaving important aspects of the manuscript unaddressed.

7. PLOS authors have the option to publish the peer review history of their article (what does this mean?). If published, this will include your full peer review and any attached files.

Reviewer #2: No

---

## [Editor Report · Acceptance letter]

27 Jan 2025

PONE-D-23-34994R3 

PLOS ONE

Dear Dr. Yi, 

I'm pleased to inform you that your manuscript has been deemed suitable for publication in PLOS ONE. Congratulations! Your manuscript is now being handed over to our production team.

Kind regards, 

on behalf of

Dr. Vinay Kumar 

Academic Editor

PLOS ONE